# Multi-class SVMs: From Tighter Data-Dependent Generalization Bounds to Novel Algorithms

**Yunwen Lei**
Department of Mathematics
City University of Hong Kong
yunwelei@cityu.edu.hk

**Ürün Dogan**
Microsoft Research
Cambridge CB1 2FB, UK
udogan@microsoft.com

**Alexander Binder**
ISTD Pillar
Singapore University of Technology and Design
Machine Learning Group, TU Berlin
alexander_binder@sutd.edu.sg

**Marius Kloft**
Department of Computer Science
Humboldt University of Berlin
kloft@hu-berlin.de

## Abstract

This paper studies the generalization performance of multi-class classification algorithms, for which we obtain—for the first time—a data-dependent generalization error bound with a *logarithmic* dependence on the class size, substantially improving the state-of-the-art linear dependence in the existing data-dependent generalization analysis. The theoretical analysis motivates us to introduce a new multi-class classification machine based on $\ell_p$-norm regularization, where the parameter $p$ controls the complexity of the corresponding bounds. We derive an efficient optimization algorithm based on Fenchel duality theory. Benchmarks on several real-world datasets show that the proposed algorithm can achieve significant accuracy gains over the state of the art.

## 1 Introduction

Typical multi-class application domains such as natural language processing [1], information retrieval [2], image annotation [3] and web advertising [4] involve tens or hundreds of thousands of classes, and yet these datasets are still growing [5]. To handle such learning tasks, it is essential to build algorithms that scale favorably with respect to the number of classes. Over the past years, much progress in this respect has been achieved on the algorithmic side [4–7], including efficient stochastic gradient optimization strategies [8].

Although also theoretical properties such as consistency [9–11] and finite-sample behavior [1, 12–15] have been studied, there still is a discrepancy between algorithms and theory in the sense that the corresponding theoretical bounds do often not scale well with respect to the number of classes. This discrepancy occurs the most strongly in research on *data-dependent* generalization bounds, that is, bounds that can measure generalization performance of prediction models purely from the training samples, and which thus are very appealing in model selection [16]. A crucial advantage of these bounds is that they can better capture the properties of the distribution that has generated the data, which can lead to tighter estimates [17] than conservative data-independent bounds.

To our best knowledge, for multi-class classification, the first data-dependent error bounds were given by [14]. These bounds exhibit a quadratic dependence on the class size and were used by [12] and [18] to derive bounds for kernel-based multi-class classification and multiple kernel learning (MKL) problems, respectively. More recently, [13] improve the quadratic dependence to a linear dependence by introducing a novel surrogate for the multi-class margin that is independent on the true realization of the class label.

However, a heavy dependence on the class size, such as linear or quadratic, implies a poor generalization guarantee for large-scale multi-class classification problems with a massive number of classes. In this paper, we show data-dependent generalization bounds for multi-class classification problems that—for the first time—exhibit a *sublinear* dependence on the number of classes. Choosing appropriate regularization, this dependence can be as mild as logarithmic. We achieve these improved bounds via the use of Gaussian complexities, while previous bounds are based on a well-known structural result on Rademacher complexities for classes induced by the maximum operator. The proposed proof technique based on Gaussian complexities exploits potential coupling among different components of the multi-class classifier, while this fact is ignored by previous analyses.

The result shows that the generalization ability is strongly impacted by the employed regularization. Which motivates us to propose a new learning machine performing block-norm regularization over the multi-class components. As a natural choice we investigate here the application of the proven $\ell_p$ norm [19]. This results in a novel $\ell_p$-norm multi-class support vector machine (MC-SVM), which contains the classical model by Crammer & Singer [20] as a special case for $p = 2$. The bounds indicate that the parameter $p$ crucially controls the complexity of the resulting prediction models.

We develop an efficient optimization algorithm for the proposed method based on its Fenchel dual representation. We empirically evaluate its effectiveness on several standard benchmarks for multi-class classification taken from various domains, where the proposed approach significantly outperforms the state-of-the-art method of [20].

The remainder of this paper is structured as follows. Section 2 introduces the problem setting and presents the main theoretical results. Motivated by which we propose a new multi-class classification model in Section 3 and give an efficient optimization algorithm based on Fenchel duality theory. In Section 4 we evaluate the approach for the application of visual image recognition and on several standard benchmark datasets taken from various application domains. Section 5 concludes.

## 2 Theory

### 2.1 Problem Setting and Notations

This paper considers multi-class classification problems with $c \geq 2$ classes. Let $\mathcal{X}$ denote the input space and $\mathcal{Y} = \{1, 2, \ldots, c\}$ denote the output space. Assume that we are given a sequence of examples $S = \{(x_1, y_1), \ldots, (x_n, y_n)\} \in (\mathcal{X} \times \mathcal{Y})^n$, independently drawn according to a probability measure $P$ defined on the sample space $\mathcal{Z} = \mathcal{X} \times \mathcal{Y}$. Based on the training examples $S$, we wish to learn a prediction rule $h$ from a space $H$ of hypotheses mapping from $\mathcal{Z}$ to $\mathbb{R}$ and use the mapping $x \to \arg\max_{y \in \mathcal{Y}} h(x, y)$ to predict (ties are broken by favoring classes with a lower index, for which our loss function defined below always counts an error). For any hypothesis $h \in H$, the margin $\rho_h(x, y)$ of the function $h$ at a labeled example $(x, y)$ is $\rho_h(x, y) := h(x, y) - \max_{y' \neq y} h(x, y')$. The prediction rule $h$ makes an error at $(x, y)$ if $\rho_h(x, y) \leq 0$ and thus the expected risk incurred from using $h$ for prediction is $R(h) := \mathbb{E}[1_{\rho_h(x,y) \leq 0}]$.

Any function $h : \mathcal{X} \times \mathcal{Y} \to \mathbb{R}$ can be equivalently represented by the vector-valued function $(h_1, \ldots, h_c)$ with $h_j(x) = h(x, j), \forall j = 1, \ldots, c$. We denote by $\widetilde{H} := \{\rho_h : h \in H\}$ the class of margin functions associated to $H$. Let $k : \mathcal{X} \times \mathcal{X} \to \mathbb{R}$ be a Mercer kernel with $\phi$ being the associated feature map, i.e., $k(x, \tilde{x}) = \langle \phi(x), \phi(\tilde{x}) \rangle$ for all $x, \tilde{x} \in \mathcal{X}$. We denote by $\| \cdot \|_*$ the dual norm of $\| \cdot \|$, i.e., $\|w\|_* := \sup_{\|\bar{w}\| \leq 1} \langle w, \bar{w} \rangle$. For a convex function $f$, we denote by $f^*$ its Fenchel conjugate, i.e., $f^*(v) := \sup_w [\langle w, v \rangle - f(w)]$. For any $\mathbf{w} = (\mathbf{w}_1, \ldots, \mathbf{w}_c)$ we define the $\ell_{2,p}$-norm by $\|\mathbf{w}\|_{2,p} := [\sum_{j=1}^c \|\mathbf{w}_j\|_2^p]^{1/p}$. For any $p \geq 1$, we denote by $p^*$ the dual exponent of $p$ satisfying $1/p + 1/p^* = 1$ and $\bar{p} := p(2 - p)^{-1}$. We require the following definitions.

**Definition 1** (Strong Convexity). *A function $f : \mathcal{X} \to \mathbb{R}$ is said to be $\beta$-strongly convex w.r.t. a norm $\| \cdot \|$ iff $\forall x, y \in \mathcal{X}$ and $\forall \alpha \in (0, 1)$, we have*

$$f(\alpha x + (1 - \alpha)y) \leq \alpha f(x) + (1 - \alpha)f(y) - \frac{\beta}{2}\alpha(1 - \alpha)\|x - y\|^2.$$

**Definition 2** (Regular Loss). *We call $\ell$ a L-regular loss if it satisfies the following properties:*

  *(i) $\ell(t)$ bounds the 0-1 loss from above: $\ell(t) \geq 1_{t \leq 0}$;*
  *(ii) $\ell$ is L-Lipschitz in the sense $|\ell(t_1) - \ell(t_2)| \leq L|t_1 - t_2|$;*

*(iii) $\ell(t)$ is decreasing and it has a zero point $c_\ell$, i.e., $\ell(c_\ell) = 0$.*

Some examples of $L$-regular loss functions include the hinge $\ell_h(t) = (1-t)_+$ and the margin loss

$$\ell_\rho(t) = 1_{t \leq 0} + (1 - t\rho^{-1})1_{0 < t \leq \rho}, \quad \rho > 0. \tag{1}$$

## 2.2 Main results

Our discussion on data-dependent generalization error bounds is based on the established methodology of Rademacher and Gaussian complexities [21].

**Definition 3** (Rademacher and Gaussian Complexity). *Let $H$ be a family of real-valued functions defined on $\mathcal{Z}$ and $S = (z_1, \ldots, z_n)$ a fixed sample of size $n$ with elements in $\mathcal{Z}$. Then, the empirical Rademacher and Gaussian complexities of $H$ with respect to the sample $S$ are defined by*

$$\mathfrak{R}_S(H) = \mathbb{E}_{\boldsymbol{\sigma}}\Big[\sup_{h \in H} \frac{1}{n} \sum_{i=1}^{n} \sigma_i h(z_i)\Big], \quad \mathfrak{G}_S(H) = \mathbb{E}_{\boldsymbol{g}}\Big[\sup_{h \in H} \frac{1}{n} \sum_{i=1}^{n} g_i h(z_i)\Big],$$

*where $\sigma_1, \ldots, \sigma_n$ are independent random variables with equal probability taking values $+1$ or $-1$, and $g_1, \ldots, g_n$ are independent $N(0,1)$ random variables.*

Note that we have the following comparison inequality relating Rademacher and Gaussian complexities (Cf. Section 4.2 in [22]):

$$\mathfrak{R}_S(H) \leq \sqrt{\frac{\pi}{2}} \mathfrak{G}_S(H) \leq 3\sqrt{\frac{\pi}{2}} \sqrt{\log n} \mathfrak{R}_S(H). \tag{2}$$

Existing work on data-dependent generalization bounds for multi-class classifiers [12–14, 18] builds on the following structural result on Rademacher complexities (e.g., [12], Lemma 8.1):

$$\mathfrak{R}_S(\max\{h_1, \ldots, h_c\} : h_j \in H_j, j = 1, \ldots, c) \leq \sum_{j=1}^{c} \mathfrak{R}_S(H_j), \tag{3}$$

where $H_1, \ldots, H_c$ are $c$ hypothesis sets. This result is crucial for the standard generalization analysis of multi-class classification since the margin $\rho_h$ involves the maximum operator, which is removed by (3), but at the expense of a linear dependency on the class size. In the following we show that this linear dependency is suboptimal because (3) does not take into account the coupling among different classes. For example, a common regularizer used in multi-class learning algorithms is $r(h) = \sum_{j=1}^{c} \|h_j\|_2^2$ [20], for which the components $h_1, \ldots, h_c$ are correlated via a $\|\cdot\|_{2,2}$ regularizer, and the bound (3) ignoring this correlation would not be effective in this case [12–14, 18].

As a remedy, we here introduce a new structural complexity result on function classes induced by general classes via the maximum operator, while allowing to preserve the correlations among different components meanwhile. Instead of considering the Rademacher complexity, Lemma 4 concerns the structural relationship of Gaussian complexities since it is based on a comparison result among different Gaussian processes.

**Lemma 4** (Structural result on Gaussian complexity). *Let $H$ be a class of functions defined on $\mathcal{X} \times \mathcal{Y}$ with $\mathcal{Y} = \{1, \ldots, c\}$. Let $g_1, \ldots, g_{nc}$ be independent $N(0,1)$ distributed random variables. Then, for any sample $S = \{x_1, \ldots, x_n\}$ of size $n$, we have*

$$\mathfrak{G}_S\big(\{\max\{h_1, \ldots, h_c\} : h = (h_1, \ldots, h_c) \in H\}\big) \leq \frac{1}{n}\mathbb{E}_{\boldsymbol{g}} \sup_{h=(h_1, \ldots, h_c) \in H} \sum_{i=1}^{n} \sum_{j=1}^{c} g_{(j-1)n+i} h_j(x_i),$$

$$\tag{4}$$

*where $\mathbb{E}_{\boldsymbol{g}}$ denotes the expectation w.r.t. to the Gaussian variables $g_1, \ldots, g_{nc}$.*

The proof of Lemma 4 is given in Supplementary Material A. Equipped with Lemma 4, we are now able to present a general data-dependent margin-based generalization bound. The proof of the following results (Theorem 5, Theorem 7 and Corollary 8) is given in Supplementary Material B.

**Theorem 5** (Data-dependent generalization bound for multi-class classification). *Let $H \subset \mathbb{R}^{\mathcal{X} \times \mathcal{Y}}$ be a hypothesis class with $\mathcal{Y} = \{1, \ldots, c\}$. Let $\ell$ be a $L$-regular loss function and denote $B_\ell := \sup_{(x,y),h} \ell(\rho_h(x,y))$. Suppose that the examples $S = \{(x_1, y_1), \ldots, (x_n, y_n)\}$ are independently*

*drawn from a probability measure defined on $\mathcal{X} \times \mathcal{Y}$. Then, for any $\delta > 0$, with probability at least $1 - \delta$, the following multi-class classification generalization bound holds for any $h \in H$:*

$$R(h) \leq \frac{1}{n} \sum_{i=1}^{n} \ell(\rho_h(x_i, y_i)) + \frac{2L\sqrt{2\pi}}{n} \mathbb{E}_{\boldsymbol{g}} \sup_{h=(h_1,\ldots,h_c)\in H} \sum_{i=1}^{n} \sum_{j=1}^{c} g_{(j-1)n+i} h_j(x_i) + 3B_\ell \sqrt{\frac{\log \frac{2}{\delta}}{2n}},$$

*where $g_1, \ldots, g_{nc}$ are independent $N(0,1)$ distributed random variables.*

**Remark 6.** Under the same condition of Theorem 5, [12] derive the following data-dependent generalization bound (Cf. Corollary 8.1 in [12]):

$$R(h) \leq \frac{1}{n} \sum_{i=1}^{n} \ell(\rho_h(x_i, y_i)) + \frac{4Lc}{n} \mathfrak{R}_S(\{x \to h(x,y) : y \in \mathcal{Y}, h \in H\}) + 3B_\ell \sqrt{\frac{\log \frac{2}{\delta}}{2n}}.$$

This linear dependence on $c$ is due to the use of (3). For comparison, Theorem 5 implies that the dependence on $c$ is governed by the term $\sum_{i=1}^{n} \sum_{j=1}^{c} g_{(j-1)n+i} h_j(x_i)$, an advantage of which is that the components $h_1, \ldots, h_c$ are jointly coupled. As we will see, this allows us to derive an improved result with a favorable dependence on $c$, when a constraint is imposed on $(h_1, \ldots, h_c)$. $\qquad \square$

The following Theorem 7 applies the general result in Theorem 5 to kernel-based methods. The hypothesis space is defined by imposing a constraint with a general strongly convex function.

**Theorem 7** (Data-dependent generalization bound for kernel-based multi-class learning algorithms and MC-SVMs)**.** *Suppose that the hypothesis space is defined by*

$$H := H_{f,\Lambda} = \{h^{\mathbf{w}} = (\langle \mathbf{w}_1, \phi(x)\rangle, \ldots, \langle \mathbf{w}_c, \phi(x)\rangle) : f(\mathbf{w}) \leq \Lambda\},$$

*where $f$ is a $\beta$-strongly convex function w.r.t. a norm $\|\cdot\|$ defined on $H$ satisfying $f^*(0) = 0$. Let $\ell$ be a $L$-regular loss function and denote $B_\ell := \sup_{(x,y),h} \ell(\rho_h(x,y))$. Let $g_1, \ldots, g_{nc}$ be independent $N(0,1)$ distributed random variables. Then, for any $\delta > 0$, with probability at least $1 - \delta$ we have*

$$R(h^{\mathbf{w}}) \leq \frac{1}{n} \sum_{i=1}^{n} \ell(\rho_{h^{\mathbf{w}}}(x_i, y_i)) + \frac{4L}{n} \sqrt{\frac{\pi\Lambda}{\beta} \mathbb{E}_{\boldsymbol{g}} \sum_{i=1}^{n} \left\| \left(g_{(j-1)n+i} \phi(x_i)\right)_{j=1,\ldots,c} \right\|_*^2} + 3B_\ell \sqrt{\frac{\log \frac{2}{\delta}}{2n}}.$$

We now consider the following specific hypothesis spaces using a $\|\cdot\|_{2,p}$ constraint:

$$H_{p,\Lambda} := \{h^{\mathbf{w}} = (\langle \mathbf{w}_1, \phi(x)\rangle, \ldots, \langle \mathbf{w}_c, \phi(x)\rangle) : \|\mathbf{w}\|_{2,p} \leq \Lambda\}, \quad 1 \leq p \leq 2. \qquad (5)$$

**Corollary 8** ($\ell_p$-norm MC-SVM generalization bound)**.** *Let $\ell$ be a $L$-regular loss function and denote $B_\ell := \sup_{(x,y),h} \ell(\rho_h(x,y))$. Then, with probability at least $1 - \delta$, for any $h^{\mathbf{w}} \in H_{p,\Lambda}$ the generalization error $R(h^{\mathbf{w}})$ can be upper bounded by:*

$$\frac{1}{n} \sum_{i=1}^{n} \ell(\rho_{h^{\mathbf{w}}}(x_i, y_i)) + 3B_\ell \sqrt{\frac{\log \frac{2}{\delta}}{2n}} + \frac{2L\Lambda}{n} \sqrt{\sum_{i=1}^{n} k(x_i, x_i)} \times \begin{cases} \sqrt{e}(4\log c)^{1+\frac{1}{2\log c}}, & \text{if } p^* \geq 2\log c, \\ (2p^*)^{1+\frac{1}{p^*}} c^{\frac{1}{p^*}}, & \text{otherwise.} \end{cases}$$

**Remark 9.** The bounds in Corollary 8 enjoy a mild dependence on the number of classes. The dependence is polynomial with exponent $1/p^*$ for $2 < p^* < 2\log c$ and becomes logarithmic if $p^* \geq 2\log c$. Even in the theoretically unfavorable case of $p = 2$ [20], the bounds still exhibit a radical dependence on the number of classes, which is substantially milder than the quadratic dependence established in [12, 14, 18] and the linear dependence established in [13]. Our generalization bound is data-dependent and shows clearly how the margin would affect the generalization performance (when $\ell$ is the margin loss $\ell_\rho$): a large margin $\rho$ would increase the empirical error while decrease the model's complexity, and vice versa. $\qquad \square$

## 2.3 Comparison of the Achieved Bounds to the State of the Art

**Related work on data-independent bounds**. The large body of theoretical work on multi-class learning considers data-independent bounds. Based on the $\ell_\infty$-norm covering number bound of linear operators, [15] obtain a generalization bound exhibiting a linear dependence on the class size, which is improved by [9] to a radical dependence of the form $O(n^{-\frac{1}{2}} (\log^{\frac{3}{2}} n) \frac{\sqrt{c}}{\rho})$. Under conditions

analogous to Corollary 8, [23] derive a class-size independent generalization guarantee. However, their bound is based on a delicate definition of margin, which is why it is commonly not used in the mainstream multi-class literature. [1] derive the following generalization bound

$$\mathbb{E}\Big[\frac{1}{p}\log\Big(1+\sum_{\tilde{y}\neq y}e^{p(\rho-\langle\hat{\mathbf{w}}_y-\hat{\mathbf{w}}_{\tilde{y}},\phi(x)\rangle)}\Big)\Big] \leq \inf_{\mathbf{w}\in H}\mathbb{E}\Big[\frac{1}{p}\log\Big(1+\sum_{\tilde{y}\neq y}e^{p(\rho-\langle\mathbf{w}_y-\mathbf{w}_{\tilde{y}},\phi(x)\rangle)}\Big)$$
$$+\frac{\lambda n}{2(n+1)}\|\mathbf{w}\|_{2,2}^2\Big]+\frac{2\sup_{x\in\mathcal{X}}k(x,x)}{\lambda n}, \quad (6)$$

where $\rho$ is a margin condition, $p > 0$ a scaling factor, and $\lambda$ a regularization parameter. Eq. (6) is class-size independent, yet Corollary 8 shows superiority in the following aspects: first, for SVMs (i.e., margin loss $\ell_\rho$), our bound consists of an empirical error ($\frac{1}{n}\sum_{i=1}^n\ell_\rho(\rho_{h^{\mathbf{w}}}(x_i,y_i))$) and a complexity term divided by the margin value (note that $L = 1/\rho$ in Corollary 8). When the margin is large (which is often desirable) [14], the last term in the bound given by Corollary 8 becomes small, while—on the contrary—-the bound (6) is an increasing function of $\rho$, which is undesirable. Secondly, Theorem 7 applies to general loss functions, expressed through a strongly convex function over a general hypothesis space, while the bound (6) only applies to a specific regularization algorithm. Lastly, all the above mentioned results are conservative data-independent estimates.

**Related work on data-dependent bounds**. The techniques used in above mentioned papers do not straightforwardly translate to data-dependent bounds, which is the type of bounds in the focus of the present work. The investigation of these was initiated, to our best knowledge, by [14]: with the structural complexity bound (3) for function classes induced via the maximal operator, [14] derive a margin bound admitting a quadratic dependency on the number of classes. [12] use these results in [14] to study the generalization performance of MC-SVMs, where the components $h_1,\ldots,h_c$ are coupled with an $\|\cdot\|_{2,p}, p\geq 1$ constraint. Due to the usage of the suboptimal Eq. (3), [12] obtain a margin bound growing quadratically w.r.t. the number of classes. [18] develop a new multi-class classification algorithm based on a natural notion called the multi-class margin of a kernel. [18] also present a novel multi-class Rademacher complexity margin bound based on Eq. (3), and the bound also depends quadratically on the class size. More recently, [13] give a refined Rademacher complexity bound with a linear dependence on the class size. The key reason for this improvement is the introduction of $\rho_{\theta,h} := \min_{y'\in\mathcal{Y}}[h(x,y)-h(x,y')+\theta 1_{y'=y}]$ bounding margin $\rho_h$ from below, and since the maximum operation in $\rho_{\theta,h}$ is applied to the set $\mathcal{Y}$ rather than the subset $\mathcal{Y}-\{y_i\}$ for $\rho_h$, one needs not to consider the random realization of $y_i$. We also use this trick in our proof of Theorem 5. However, [13] fail to improve this linear dependence to a logarithmic dependence, as we achieved in Corollary 8, due to the use of the suboptimal structural result (3).

## 3 Algorithms

Motivated by the generalization analysis given in Section 2, we now present a new multi-class learning algorithm, based on performing empirical risk minimization in the hypothesis space (5). This corresponds to the following $\ell_p$-norm MC-SVM ($1 \leq p \leq 2$):

**Problem 10** (Primal problem: $\ell_p$-norm MC-SVM)**.**

$$\min_{\mathbf{w}}\frac{1}{2}\Big[\sum_{j=1}^c\|\mathbf{w}_j\|_2^p\Big]^{\frac{2}{p}}+C\sum_{i=1}^n\ell(t_i),$$
$$s.t.\ t_i=\langle\mathbf{w}_{y_i},\phi(x_i)\rangle-\max_{y\neq y_i}\langle\mathbf{w}_y,\phi(x_i)\rangle, \quad (P)$$

For $p = 2$ we recover the seminal multi-class algorithm by Crammer & Singer [20] (CS), which is thus a special case of the proposed formulation. An advantage of the proposed approach over [20] can be that, as shown in Corollary 8, the dependence of the generalization performance on the class size becomes milder as $p$ decreases to $1$.

### 3.1 Dual problems

Since the optimization problem (P) is convex, we can derive the associated dual problem for the construction of efficient optimization algorithms. The derivation of the following dual problem is deferred to Supplementary Material C. For a matrix $\boldsymbol{\alpha}\in\mathbb{R}^{n\times c}$, we denote by $\boldsymbol{\alpha}_i$ the $i$-th row. Denote by $e_j$ the $j$-th unit vector in $\mathbb{R}^c$ and $\mathbf{1}$ the vector in $\mathbb{R}^c$ with all components being zero.

**Problem 11** (Completely dualized problem for general loss). *The Lagrangian dual of* (10) *is:*

$$\sup_{\boldsymbol{\alpha} \in \mathbb{R}^{n \times c}} -\frac{1}{2} \Big[ \sum_{j=1}^{c} \| \sum_{i=1}^{n} \alpha_{ij} \phi(x_i) \|_2^{\frac{p}{p-1}} \Big]^{\frac{2(p-1)}{p}} - C \sum_{i=1}^{n} \ell^*(-\frac{\alpha_{iy_i}}{C}) \tag{D}$$

$$s.t. \; \alpha_{ij} \leq 0 \; \wedge \; \boldsymbol{\alpha}_i \cdot \mathbf{1} = 0, \quad \forall j \neq y_i, i = 1, \ldots, n.$$

**Theorem 12** (REPRESENTER THEOREM). *For any dual variable* $\boldsymbol{\alpha} \in \mathbb{R}^{n \times c}$, *the associated primal variable* $\mathbf{w} = (\mathbf{w}_1, \ldots, \mathbf{w}_c)$ *minimizing the Lagrangian saddle problem can be represented by:*

$$\mathbf{w}_j = \Big[ \sum_{\tilde{j}=1}^{c} \| \sum_{i=1}^{n} \alpha_{i\tilde{j}} \phi(x_i) \|_2^{p^*} \Big]^{\frac{2}{p^*}-1} \| \sum_{i=1}^{n} \alpha_{ij} \phi(x_i) \|_2^{p^*-2} \Big[ \sum_{i=1}^{n} \alpha_{ij} \phi(x_i) \Big].$$

For the hinge loss $\ell_h(t) = (1-t)_+$, we know its Fenchel-Legendre conjugate is $\ell_h^*(t) = t$ if $-1 \leq t \leq 0$ and $\infty$ elsewise. Hence $\ell_h^*(-\frac{\alpha_{iy_i}}{C}) = -\frac{\alpha_{iy_i}}{C}$ if $-1 \leq -\frac{\alpha_{iy_i}}{C} \leq 0$ and $\infty$ elsewise. Now we have the following dual problem for the hinge loss function:

**Problem 13** (Completely dualized problem for the hinge loss ($\ell_p$-norm MC-SVM)).

$$\sup_{\boldsymbol{\alpha} \in \mathbb{R}^{n \times c}} -\frac{1}{2} \Big[ \sum_{j=1}^{c} \| \sum_{i=1}^{n} \alpha_{ij} \phi(x_i) \|_2^{\frac{p}{p-1}} \Big]^{\frac{2(p-1)}{p}} + \sum_{i=1}^{n} \alpha_{iy_i} \tag{7}$$

$$s.t. \; \boldsymbol{\alpha}_i \leq \mathbf{e}_{y_i} \cdot C \; \wedge \; \boldsymbol{\alpha}_i \cdot \mathbf{1} = 0, \quad \forall i = 1, \ldots, n.$$

### 3.2 Optimization Algorithms

The dual problems (D) and (7) are not quadratic programs for $p \neq 2$, and thus generally not easy to solve. To circumvent this difficulty, we rewrite Problem 10 as the following equivalent problem:

$$\min_{\mathbf{w}, \boldsymbol{\beta}} \sum_{j=1}^{c} \frac{\|\mathbf{w}_j\|_2^2}{2\beta_j} + C \sum_{i=1}^{n} \ell(t_i)$$

$$s.t. \; t_i \leq \langle \mathbf{w}_{y_i}, \phi(x_i) \rangle - \langle \mathbf{w}_y, \phi(x_i) \rangle, \quad y \neq y_i, i = 1, \ldots, n, \tag{8}$$

$$\|\boldsymbol{\beta}\|_{\bar{p}} \leq 1, \bar{p} = p(2-p)^{-1}, \beta_j \geq 0.$$

The class weights $\beta_1, \ldots, \beta_c$ in Eq. (8) play a similar role as the kernel weights in $\ell_p$-norm MKL algorithms [19]. The equivalence between problem (P) and Eq. (8) follows directly from Lemma 26 in [24], which shows that the optimal $\boldsymbol{\beta} = (\beta_1, \ldots, \beta_c)$ in Eq. (8) can be explicitly represented in closed form. Motivated by the recent work on $\ell_p$-norm MKL, we propose to solve the problem (8) via alternately optimizing $\mathbf{w}$ and $\boldsymbol{\beta}$. As we will show, given temporarily fixed $\boldsymbol{\beta}$, the optimization of $\mathbf{w}$ reduces to a standard multi-class classification problem. Furthermore, the update of $\boldsymbol{\beta}$, given fixed $\mathbf{w}$, can be achieved via an analytic formula.

**Problem 14** (Partially dualized problem for a general loss). *For fixed* $\boldsymbol{\beta}$, *the partial dual problem for the sub-optimization problem* (8) *w.r.t.* $\mathbf{w}$ *is*

$$\sup_{\boldsymbol{\alpha} \in \mathbb{R}^{n \times c}} -\frac{1}{2} \sum_{j=1}^{c} \beta_j \| \sum_{i=1}^{n} \alpha_{ij} \phi(x_i) \|_2^2 - C \sum_{i=1}^{n} \ell^*(-\frac{\alpha_{iy_i}}{C}) \tag{9}$$

$$s.t. \; \alpha_{ij} \leq 0 \; \wedge \; \boldsymbol{\alpha}_i \cdot \mathbf{1} = 0, \quad \forall j \neq y_i, i = 1, \ldots, n.$$

*The primal variable* $\mathbf{w}$ *minimizing the associated Lagrangian saddle problem is*

$$\mathbf{w}_j = \beta_j \sum_{i=1}^{n} \alpha_{ij} \phi(x_i). \tag{10}$$

We defer the proof to Supplementary Material C. Analogous to Problem 13, we have the following partial dual problem for the hinge loss.

**Problem 15** (Partially dualized problem for the hinge loss ($\ell_p$-norm MC-SVM)).

$$\sup_{\boldsymbol{\alpha} \in \mathbb{R}^{n \times c}} f(\boldsymbol{\alpha}) := -\frac{1}{2} \sum_{j=1}^{c} \beta_j \| \sum_{i=1}^{n} \alpha_{ij} \phi(x_i) \|_2^2 + \sum_{i=1}^{n} \alpha_{iy_i} \tag{11}$$

$$s.t. \; \boldsymbol{\alpha}_i \leq \mathbf{e}_{y_i} \cdot C \; \wedge \; \boldsymbol{\alpha}_i \cdot \mathbf{1} = 0, \quad \forall i = 1, \ldots, n.$$

The Problems 14 and 15 are quadratic, so we can use the dual coordinate ascent algorithm [25] to very efficiently solve them for the case of linear kernels. To this end, we need to compute the gradient and solve the restricted problem of optimizing only one $\alpha_i, \forall i$, keeping all other dual variables fixed [25]. The gradient of $f$ can be exactly represented by $\mathbf{w}$:

$$\frac{\partial f}{\partial \alpha_{ij}} = -\beta_j \sum_{\tilde{i}=1}^{n} \alpha_{\tilde{i}j} k(x_i, x_{\tilde{i}}) + 1_{y_i=j} = 1_{y_i=j} - \langle \mathbf{w}_j, \phi(x_i) \rangle. \tag{12}$$

Suppose the additive change to be applied to the current $\alpha_i$ is $\delta\alpha_i$, then from (12) we have

$$f(\alpha_1, \ldots, \alpha_{i-1}, \alpha_i + \delta\alpha_i, \alpha_{i+1}, \ldots, \alpha_n) = \sum_{j=1}^{c} \frac{\partial f}{\partial \alpha_{ij}} \delta\alpha_{ij} - \frac{1}{2} \sum_{j=1}^{c} \beta_j k(x_i, x_i)[\delta\alpha_{ij}]^2 + \text{const.}$$

Therefore, the sub-problem of optimizing $\boldsymbol{\delta}\alpha_i$ is given by

$$\max_{\boldsymbol{\delta\alpha_i}} \; -\frac{1}{2} \sum_{j=1}^{c} \beta_j k(x_i, x_i)[\delta\alpha_{ij}]^2 + \sum_{j=1}^{c} \frac{\partial f}{\partial \alpha_{ij}} \delta\alpha_{ij} \tag{13}$$

$$\text{s.t. } \boldsymbol{\delta\alpha_i} \leq \boldsymbol{e}_{y_i} \cdot C - \boldsymbol{\alpha}_i \; \wedge \; \boldsymbol{\delta\alpha_i} \cdot \mathbf{1} = 0.$$

We now consider the subproblem of updating class weights $\boldsymbol{\beta}$ with temporarily fixed $\boldsymbol{w}$, for which we have the following analytic solution. The proof is deferred to the Supplementary Material C.1.

**Proposition 16.** *(Solving the subproblem with respect to the class weights) Given fixed* $\mathbf{w}_j$, *the minimal* $\beta_j$ *optimizing the problem* (8) *is attained at*

$$\beta_j = \|\mathbf{w}_j\|_2^{2-p} \left( \sum_{\tilde{j}=1}^{c} \|\mathbf{w}_{\tilde{j}}\|_2^p \right)^{\frac{p-2}{p}}. \tag{14}$$

The update of $\beta_j$ based on Eq. (14) requires calculating $\|\mathbf{w}_j\|_2^2$, which can be easily fulfilled by recalling the representation established in Eq. (10).

The resulting training algorithm for the proposed $\ell_p$-norm MC-SVM is given in Algorithm 1. The algorithm alternates between solving a MC-SVM problem for fixed class weights (Line 3) and updating the class weights in a closed-form manner (Line 5). Recall that Problem 11 establishes a completely dualized problem, which can be used as a sound stopping criterion for Algorithm 1.

---

**Algorithm 1:** Training algorithm for $\ell_p$-norm MC-SVM.

---

**input**: examples $\{(x_i, y_i)_{i=1}^{n}\}$ and the kernel $k$.

initialize $\beta_j = \sqrt[\bar{p}]{1/c}, \mathbf{w}_j = 0$ for all $j = 1, \ldots, c$
**while** *Optimality conditions are not satisfied* **do**
  optimize the multi-class classification problem (9)
  compute $\|\mathbf{w}_j\|_2^2$ for all $j = 1, \ldots, c$, according to Eq. (10)
  update $\beta_j$ for all $j = 1, \ldots, c$, according to Eq. (14)
**end**

---

## 4   Empirical Analysis

We implemented the proposed $\ell_p$-norm MC-SVM algorithm (Algorithm 1) in C++ and solved the involved MC-SVM problem using dual coordinate ascent [25]. We experiment on six benchmark datasets: the Sector dataset studied in [26], the News 20 dataset collected by [27], the Rcv1 dataset collected by [28], the Birds 15, Birds 50 as a part from [29] and the Caltech 256 collected by griffin2007caltech. We used fc6 features from the BVLC reference caffenet from [30]. Table 1 gives an information on these datasets.

We compare with the classical CS in [20], which constitutes a strong baseline for these datasets [25]. We employ a 5-fold cross validation on the training set to tune the regularization parameter $C$ by grid search over the set $\{2^{-12}, 2^{-11}, \ldots, 2^{12}\}$ and $p$ from 1.1 to 2 with 10 equidistant points. We repeat the experiments 10 times, and report in Table 2 on the average accuracy and standard deviations attained on the test set.

| Dataset | No. of Classes | No. of Training Examples | No. of Test Examples | No. of Attributes |
|---|---|---|---|---|
| Sector | 105 | $6,412$ | $3,207$ | $55,197$ |
| News 20 | 20 | $15,935$ | $3,993$ | $62,060$ |
| Rcv1 | 53 | $15,564$ | $518,571$ | $47,236$ |
| Birds 15 | 200 | $3,000$ | $8,788$ | $4,096$ |
| Birds 50 | 200 | $9,958$ | $1,830$ | $4,096$ |
| Caltech 256 | 256 | $12,800$ | $16,980$ | $4,096$ |

Table 1: Description of datasets used in the experiments.

| Method / Dataset | Sector | News 20 | Rcv1 | Birds 15 | Birds 50 | Caltech 256 |
|---|---|---|---|---|---|---|
| $\ell_p$-norm MC-SVM | **94.20±0.34** | **86.19±0.12** | **85.74±0.71** | **13.73±1.4** | **27.86±0.2** | **56.00±1.2** |
| Crammer & Singer | 93.89±0.27 | 85.12±0.29 | 85.21±0.32 | 12.53±1.6 | 26.28±0.3 | 54.96±1.1 |

Table 2: Accuracies achieved by CS and the proposed $\ell_p$-norm MC-SVM on the benchmark datasets.

We observe that the proposed $\ell_p$-norm MC-SVM consistently outperforms CS [20] on all considered datasets. Specifically, our method attains $0.31\%$ accuracy gain on Sector, $1.07\%$ accuracy gain on News 20, $0.53\%$ accuracy gain on Rcv1, $1.2\%$ accuracy gain on Birds 15, $1.58\%$ accuracy gain on Birds 50, and $1.04\%$ accuracy gain on Birds 15. We perform a Wilcoxon signed rank test between the accuracies of CS and our method on the benchmark datasets, and the p-value is $0.03$, which means our method is significantly better than CS at the significance level of $0.05$. These promising results indicate that the proposed $\ell_p$-norm MC-SVM could further lift the state of the art in multi-class classification, even in real-world applications beyond the ones studied in this paper.

## 5 Conclusion

Motivated by the ever growing size of multi-class datasets in real-world applications such as image annotation and web advertising, which involve tens or hundreds of thousands of classes, we studied the influence of the class size on the generalization behavior of multi-class classifiers. We focus here on data-dependent generalization bounds enjoying the ability to capture the properties of the distribution that has generated the data. Of independent interest, for hypothesis classes that are given as a maximum over base classes, we developed a new structural result on Gaussian complexities that is able to preserve the coupling among different components, while the existing structural results ignore this coupling and may yield suboptimal generalization bounds. We applied the new structural result to study learning rates for multi-class classifiers, and derived, for the first time, a data-dependent bound with a logarithmic dependence on the class size, which substantially outperforms the linear dependence in the state-of-the-art data-dependent generalization bounds.

Motivated by the theoretical analysis, we proposed a novel $\ell_p$-norm MC-SVM, where the parameter $p$ controls the complexity of the corresponding bounds. This class of algorithms contains the classical CS [20] as a special case for $p = 2$. We developed an effective optimization algorithm based on the Fenchel dual representation. For several standard benchmarks taken from various domains, the proposed approach surpassed the state-of-the-art method of CS [20] by up to $1.5\%$.

A future direction will be to derive a data-dependent bound that is completely independent of the class size (even overcoming the mild logarithmic dependence here). To this end, we will study more powerful structural results than Lemma 4 for controlling complexities of function classes induced via the maximum operator. As a good starting point, we will consider $\ell_\infty$-norm covering numbers.

**Acknowledgments**

We thank Mehryar Mohri for helpful discussions. This work was partly funded by the German Research Foundation (DFG) award KL 2698/2-1.

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
