[Supplementary Material · appendix.pdf]

# Supplementary Material

## A  Proofs of Structural Results on Gaussian Complexities

Our discussion on complexity bound is based on the following comparison result of Gaussian processes due to [1].

**Lemma A.1** (E.g., Theorem 1 in [2]). *Let $\{\mathfrak{X}_\theta : \theta \in \Theta\}$ and $\{\mathfrak{Y}_\theta : \theta \in \Theta\}$ be two mean-zero separable Gaussian processes indexed by the same set $\Theta$ and suppose that*

$$\mathbb{E}[(\mathfrak{X}_\theta - \mathfrak{X}_{\bar{\theta}})^2] \le \mathbb{E}[(\mathfrak{Y}_\theta - \mathfrak{Y}_{\bar{\theta}})^2], \quad \forall \theta, \bar{\theta} \in \Theta. \tag{A.1}$$

*Then,*

$$\mathbb{E}[\sup_{\theta \in \Theta} \mathfrak{X}_\theta] \le \mathbb{E}[\sup_{\theta \in \Theta} \mathfrak{Y}_\theta].$$

**Proof of Lemma 4**. Define two mean-zero separable Gaussian processes indexed by the finite dimensional Euclidean space $\{(h(x_1), \ldots, h(x_n)) : h = (h_1, \ldots, h_c) \in H\}$ (for simplicity, we use here the index $h$ to denote $(h(x_1), \ldots, h(x_n))$)

$$\mathfrak{X}_h := \sum_{i=1}^n g_i \max\{h_1(x_i), h_2(x_i), \ldots, h_c(x_i)\},$$

$$\mathfrak{Y}_h := \sum_{i=1}^n \sum_{j=1}^c g_{(j-1)n+i} h_j(x_i), \qquad \forall h \in H.$$

For any $h = (h_1, \ldots, h_c), \bar{h} = (\bar{h}_1, \ldots, \bar{h}_c) \in H$, the independence of the $g_i$ and the equalities $\mathbb{E}g_i^2 = 1$ imply that

$$\mathbb{E}[(\mathfrak{X}_h - \mathfrak{X}_{\bar{h}})^2] = \sum_{i=1}^n \big[\max\{h_1(x_i), \ldots, h_c(x_i)\} - \max\{\bar{h}_1(x_i), \ldots, \bar{h}_c(x_i)\}\big]^2$$

$$\mathbb{E}[(\mathfrak{Y}_h - \mathfrak{Y}_{\bar{h}})^2] = \sum_{i=1}^n \sum_{j=1}^c |h_j(x_i) - \bar{h}_j(x_i)|^2. \tag{A.2}$$

For any $\boldsymbol{a} = (a_1, \ldots, a_c), \boldsymbol{b} = (b_1, \ldots, b_c) \in \mathbb{R}^c$, it can be directly checked that

$$|\max\{a_1, \ldots, a_c\} - \max\{b_1, \ldots, b_c\}| \le \max\{|a_1 - b_1|, \ldots, |a_c - b_c|\} \le \sum_{i=1}^c |a_i - b_i|. \tag{A.3}$$

Applying the above inequality with $\boldsymbol{a} = (h_1(x_i), \ldots, h_c(x_i)), \boldsymbol{b} = (\bar{h}_1(x_i), \ldots, \bar{h}_c(x_i)), i = 1, \ldots, n$, yields directly the following bounds relating the increments of the two Gaussian processes $\mathfrak{X}_h, \mathfrak{Y}_h$:

$$\mathbb{E}[(\mathfrak{X}_h - \mathfrak{X}_{\bar{h}})^2] \overset{(A.2)}{=} \sum_{i=1}^n \big[\max\{h_1(x_i), \ldots, h_c(x_i)\} - \max\{\bar{h}_1(x_i), \ldots, \bar{h}_c(x_i)\}\big]^2$$

$$\overset{(A.3)}{\le} \sum_{i=1}^n \max\{|h_1(x_i) - \bar{h}_1(x_i)|, \ldots, |h_c(x_i) - \bar{h}_c(x_i)|\}^2$$

$$= \sum_{i=1}^n \max\{|h_1(x_i) - \bar{h}_1(x_i)|^2, \ldots, |h_c(x_i) - \bar{h}_c(x_i)|^2\}$$

$$\overset{(A.3)}{\le} \sum_{i=1}^n \sum_{j=1}^c |h_j(x_i) - \bar{h}_j(x_i)|^2 \overset{(A.2)}{=} \mathbb{E}[(\mathfrak{Y}_h - \mathfrak{Y}_{\bar{h}})^2], \quad \forall h, \bar{h} \in H.$$

That is, the condition (A.1) holds and therefore Lemma A.1 can be applied here to yield the stated result. $\square$

The following structural lemma regarding the Gaussian complexity of simplistic multi-class hypothesis spaces (not involving any argmax operator) will be used further below in the proof of Theorem 5.

**Lemma A.2.** *Let $H$ be a class of functions defined on $\mathcal{X} \times \mathcal{Y}$ with $\mathcal{Y} = \{1, \ldots, c\}$. Let $S = \{(x_1, y_1), \ldots, (x_n, y_n)\}$ be a sequence of examples. Let $g_1, \ldots, g_{nc}$ be independent $N(0, 1)$ distributed random variables. Then the empirical Gaussian complexity of $H$ can be controlled by:*

$$\mathfrak{G}_S(H) \leq \frac{1}{n} \mathbb{E}_{\boldsymbol{g}} \sup_{h=(h_1,\ldots,h_c)\in H} \sum_{i=1}^{n} \sum_{j=1}^{c} g_{(j-1)n+i} h_j(x_i).$$

*Proof.* Define two Gaussian processes indexed by $H$:

$$\mathfrak{X}_h := \sum_{i=1}^{n} g_i h_{y_i}(x_i), \quad \mathfrak{Y}_h := \sum_{i=1}^{n} \sum_{j=1}^{c} g_{(j-1)n+i} h_j(x_i), \quad \forall h \in H.$$

For any $h, \bar{h} \in H$, it is obvious that

$$\mathbb{E}[(\mathfrak{X}_h - \mathfrak{X}_{\bar{h}})^2] = \sum_{i=1}^{n} [h_{y_i}(x_i) - \bar{h}_{y_i}(x_i)]^2$$

$$\leq \sum_{i=1}^{n} \left[ (h_1(x_i) - \bar{h}_1(x_i))^2 + \cdots + (h_c(x_i) - \bar{h}_c(x_i))^2 \right]$$

$$= \mathbb{E}[(\mathfrak{Y}_h - \mathfrak{Y}_{\bar{h}})^2].$$

Now the stated inequality follows directly from Lemma A.1. $\qquad\square$

# B  Proof of Generalization Bounds for Multi-class Classification

## B.1  Proof of Generalization Bound for General Multi-Class Classification (Theorem 5)

One of the main results of this paper is proved in this section. We first give a concentration inequality attributed to [3].

**Lemma B.1** (McDiarmid inequality [3]). *Let $Z_1, \ldots, Z_n$ be independent random variables taking values in a set $\mathcal{Z}$, and assume that $f : \mathcal{Z}^n \to \mathbb{R}$ satisfies*

$$\sup_{\substack{z_1,\ldots,z_n \\ \bar{z}_k \in \mathcal{Z}}} |f(z_1, \cdots, z_n) - f(z_1, \cdots, z_{k-1}, \bar{z}_k, z_{k+1}, \cdots, z_n)| \leq c_i \tag{B.1}$$

*for $1 \leq k \leq n$. Then, for any $0 < \delta < 1$, with probability at least $1 - \delta$ we have*

$$f(Z_1, \ldots, Z_n) \leq \mathbb{E}f(Z_1, \ldots, Z_n) + \sqrt{\frac{\sum_{k=1}^{n} c_k^2 \log(1/\delta)}{2}}.$$

**Proof of Theorem 5**. For any $\theta > 0$, introduce the following function bounding $\rho_h(x, y)$ from below:

$$\rho_{\theta,h}(x, y) = h(x, y) - \max_{y'\in\mathcal{Y}}[h(x, y') - \theta 1_{y'=y}] = \min_{y'\in\mathcal{Y}}[h(x, y) - h(x, y') + \theta 1_{y'=y}].$$

It can be checked that $\rho_{\theta,h}(x, y) = \min(\rho_h(x, y), \theta)$. Introduce two function classes derived from $\rho_{\theta,h}$:

$$\widetilde{H_\theta} = \{\rho_{\theta,h}(x, y) : h \in H\}, \qquad \ell \circ \widetilde{H_\theta} = \{\ell(\rho_{\theta,h}(x, y)) : h \in H\}.$$

According to the definition of $L$-regular loss function and the relationship $\rho_{\theta,h} \leq \rho_h$, we have

$$R(h) = \mathbb{E}[1_{\rho_h(X,Y)} \leq 0] \leq \mathbb{E}[1_{\rho_{\theta,h}(X,Y)} \leq 0] \leq \mathbb{E}[\ell(\rho_{\theta,h}(X, Y))],$$

which, together with McDiarmid inequality [3] and the symmetrization technique (e.g., Theorem 4.4 in [4]), yields the following inequality

$$R(h) \leq \frac{1}{n} \sum_{i=1}^{n} \ell(\rho_{\theta,h}(x_i, y_i)) + 2\mathfrak{R}_S(\ell \circ \widetilde{H_\theta}) + 3B_\ell \sqrt{\frac{\log\frac{2}{\delta}}{2n}}, \quad \forall h \in H \tag{B.2}$$

with probability at least $1 - \delta$.

For the fixed parameter $\theta = c_\ell$, we observe that $\rho_{\theta,h}(x,y) = \min(\rho_h(x,y), c_\ell)$. If $\rho_h(x,y) > c_\ell$, the definition of $L$-regular loss implies that

$$\ell(\rho_{\theta,h}(x,y)) = \ell(c_\ell) = 0 = \ell(\rho_h(x,y)).$$

Otherwise, we have $\rho_{\theta,h}(x,y) = \rho_h(x,y)$. Therefore, for any $(x,y)$ we have $\ell(\rho_{\theta,h}(x,y)) = \ell(\rho_h(x,y))$, which, coupled with the Lipschitz property of $\ell$ and Eq. (B.2), yields the following inequality with probability at least $1 - \delta$:

$$R(h) \leq \frac{1}{n} \sum_{i=1}^{n} \ell(\rho_h(x_i, y_i)) + 2L\mathfrak{R}_S(\widetilde{H_\theta}) + 3B_\ell \sqrt{\frac{\log \frac{2}{\delta}}{2n}}, \quad \forall h \in H. \tag{B.3}$$

The Rademacher complexity of $\widetilde{H_\theta}$ satisfies the following inequality:

$$\mathfrak{R}_S(\widetilde{H_\theta}) = \frac{1}{n} \mathbb{E}_\sigma \Big[ \sup_{h \in H} \sum_{i=1}^{n} \sigma_i \big( h(x_i, y_i) - \max_{y \in \mathcal{Y}} (h(x_i, y) - \theta 1_{y=y_i}) \big) \Big]$$

$$\leq \frac{1}{n} \mathbb{E}_\sigma [\sup_{h \in H} \sum_{i=1}^{n} \sigma_i h(x_i, y_i)] + \frac{1}{n} \mathbb{E}_\sigma \Big[ \sup_{h \in H} \sum_{i=1}^{n} \sigma_i \max_{y \in \mathcal{Y}} (h(x_i, y) - \theta 1_{y=y_i}) \Big]$$

$$\leq \sqrt{\frac{\pi}{2}} \mathfrak{G}_S(H) + \frac{1}{n} \sqrt{\frac{\pi}{2}} \mathbb{E}_g \Big[ \sup_{h=(h_1,\ldots,h_c) \in H} \sum_{i=1}^{n} g_i \max(h_1(x_i) - \theta 1_{y_i=1}, \ldots, h_c(x_i) - \theta 1_{y_i=c}) \Big],$$
$$\tag{B.4}$$

where the last step follows from the relationship between Gaussian and Rademacher processes expressed in Eq. (2). Furthermore, according to Lemma 4, the last term of the above inequality can be addressed by

$$\mathbb{E}_{\boldsymbol{g}} \Big[ \sup_{h=(h_1,\ldots,h_c) \in H} \sum_{i=1}^{n} g_i \max\{ h_1(x_i) - \theta 1_{y_i=1}, \ldots, h_c(x_i) - \theta 1_{y_i=c} \} \Big]$$

$$\overset{\text{Lemma 4}}{\leq} \mathbb{E}_{\boldsymbol{g}} \sup_{h=(h_1,\ldots,h_c) \in H} \sum_{i=1}^{n} \sum_{j=1}^{c} g_{(j-1)n+i} (h_j(x_i) - \theta 1_{y_i=j})$$

$$= \mathbb{E}_{\boldsymbol{g}} \sup_{h=(h_1,\ldots,h_c) \in H} \sum_{i=1}^{n} \sum_{j=1}^{c} g_{(j-1)n+i} h_j(x_i) - \underbrace{\mathbb{E}_{\boldsymbol{g}} \sum_{i=1}^{n} \sum_{j=1}^{c} g_{(j-1)n+i} \theta 1_{y_i=j}}_{=0}$$

$$= \mathbb{E}_{\boldsymbol{g}} \sup_{h=(h_1,\ldots,h_c) \in H} \sum_{i=1}^{n} \sum_{j=1}^{c} g_{(j-1)n+i} h_j(x_i).$$

With this inequality and using Lemma A.2 to tackle $\mathfrak{G}_S(H)$, we immediately derive the following bound on $\mathfrak{R}_S(\widetilde{H_\theta})$:

$$\mathfrak{R}_S(\widetilde{H_\theta}) \leq \frac{\sqrt{2\pi}}{n} \mathbb{E}_{\boldsymbol{g}} \sup_{h=(h_1,\ldots,h_c) \in H} \sum_{i=1}^{n} \sum_{j=1}^{c} g_{(j-1)n+i} h_j(x_i).$$

Plugging this Rademacher complexity bound back into Eq. (B.3), we obtain the stated result. $\square$

## B.2 Proof of Generalization Bound for Kernel-Based Multi-Class Classification and MC-SVMs (Theorem 7)

To apply Theorem 5, we need to control the term $\sup_{h \in H} \sum_{i=1}^{n} \sum_{j=1}^{c} g_{(j-1)n+i} h_j(x_i)$, which we tackle by the following lemma due to [5].

**Lemma B.2** (Corollary 4 in [5]). *If $f$ is $\beta$-strongly convex w.r.t. $\|\cdot\|$ and $f^*(\mathbf{0}) = 0$, then, for any sequence $v_1, \ldots, v_n$ and for any $\mu$ we have*

$$\sum_{i=1}^{n} \langle v_i, \mu \rangle - f(\mu) \leq \sum_{i=1}^{n} \langle \nabla f^*(v_{1:i-1}), v_i \rangle + \frac{1}{2\beta} \sum_{i=1}^{n} \|v_i\|_*^2,$$

*where $v_{1:i}$ denotes the sum $\sum_{j=1}^{i} v_j$.*

**Proof of Theorem 7.** For the hypothesis space $H$ and any $\lambda > 0$, applying Lemma B.2 with $\mu = (\mathbf{w}_1, \ldots, \mathbf{w}_c)$ and $v_i = \lambda(g_i\phi(x_i), g_{n+i}\phi(x_i), \ldots, g_{(c-1)n+i}\phi(x_i))$, we have

$$\lambda \sup_{h^{\mathbf{w}} \in H} \sum_{i=1}^{n} \sum_{j=1}^{c} g_{(j-1)n+i} h_j^{\mathbf{w}}(x_i) = \sup_{h^{\mathbf{w}} \in H} \sum_{i=1}^{n} \sum_{j=1}^{c} g_{(j-1)n+i} \langle \mathbf{w}_j, \lambda\phi(x_i) \rangle$$

$$= \sup_{h^{\mathbf{w}} \in H} \sum_{i=1}^{n} \langle (\mathbf{w}_1, \ldots, \mathbf{w}_c), (\lambda g_i\phi(x_i), \lambda g_{n+i}\phi(x_i), \ldots, \lambda g_{(c-1)n+i}\phi(x_i)) \rangle$$

$$\leq \sup_{h^{\mathbf{w}} \in H} f(\mathbf{w}_1, \ldots, \mathbf{w}_c) + \sum_{i=1}^{n} \langle \nabla f^*(v_{1:i-1}), v_i \rangle + \frac{\lambda^2}{2\beta} \sum_{i=1}^{n} \|(g_i\phi(x_i), g_{n+i}\phi(x_i), \ldots, g_{(c-1)n+i}\phi(x_i))\|_*^2.$$

Taking expectation on both sides w.r.t. the Gaussian variables $g_1, \ldots, g_{nc}$, the term $\sum_{i=1}^{n} \langle \nabla f^*(v_{1:i-1}), v_i \rangle$ vanishes, and therefore we obtain

$$\mathbb{E}_{\boldsymbol{g}} \sup_{h^{\mathbf{w}} \in H} \sum_{i=1}^{n} \sum_{j=1}^{c} g_{(j-1)n+i} h_j^{\mathbf{w}}(x_i) \leq \frac{\Lambda}{\lambda} + \frac{\lambda}{2\beta} \sum_{i=1}^{n} \mathbb{E}_{\boldsymbol{g}} \|(g_i\phi(x_i), g_{n+i}\phi(x_i), \ldots, g_{(c-1)n+i}\phi(x_i))\|_*^2.$$

Choosing $\lambda = \sqrt{\frac{2\beta\Lambda}{\sum_{i=1}^{n} \mathbb{E}_{\boldsymbol{g}} \|(g_i\phi(x_i), g_{n+i}\phi(x_i), \ldots, g_{(c-1)n+i}\phi(x_i))\|_*^2}}$, the above inequality translates to

$$\mathbb{E}_{\boldsymbol{g}} \sup_{h^{\mathbf{w}} \in H} \sum_{i=1}^{n} \sum_{j=1}^{c} g_{(j-1)n+i} h_j^{\mathbf{w}}(x_i) \leq \sqrt{\frac{2\Lambda}{\beta} \sum_{i=1}^{n} \mathbb{E}_{\boldsymbol{g}} \|(g_i\phi(x_i), g_{n+i}\phi(x_i), \ldots, g_{(c-1)n+i}\phi(x_i))\|_*^2}.$$

Putting the above complexity bound into Theorem 5, we obtain the stated result. $\qquad\square$

## B.3 Proof of Generalization Bound for $\ell_p$-norm Multi-class SVMs (Corollary 8)

The following simple lemma controls the $p$-th moment of a $N(0,1)$ distributed random variable. We give the proof here for completeness.

**Lemma B.3.** *Let $g$ be $N(0,1)$ distributed. For any $p > 0$, the $p$-th moment of $g$ can be bounded by*

$$[\mathbb{E}|g|^p]^{\frac{1}{p}} \leq (2p)^{\frac{1}{2} + \frac{1}{p}}.$$

*Proof.* Let $\forall n \in \mathbb{N}_+ : \Gamma(n) = (n-1)!$ be the Gamma function. The $p$-th moment of a $N(0,1)$ distributed random variable can be exactly expressed via Gamma function [6]:

$$\mathbb{E}|g|^p = \frac{2^{\frac{p}{2}}}{\sqrt{\pi}} \Gamma\left(\frac{p+1}{2}\right) \leq \frac{2^{\frac{p}{2}}}{\sqrt{\pi}} \Gamma\left(\lceil\frac{p+1}{2}\rceil\right)$$

$$= \frac{2^{\frac{p}{2}}}{\sqrt{\pi}} \lceil\frac{p-1}{2}\rceil! \leq \frac{2^{\frac{p}{2}}}{\sqrt{\pi}} \sqrt{2\pi} \lceil\frac{p-1}{2}\rceil^{\lceil\frac{p-1}{2}\rceil + \frac{1}{2}}$$

$$\leq (2p)^{\frac{p}{2}+1},$$

where in the above deduction we have used Stirling's approximation [7]:

$$n! \leq \sqrt{2\pi} n^{n+\frac{1}{2}} e^{-n+1/(12n)}.$$

$$\square$$

**Proof of Corollary** 8. Let $g_1, \ldots, g_{nc}$ be independent $N(0,1)$ distributed random variables. Denote by $\tau_s = [\mathbb{E}|g_1|^s]^{\frac{1}{s}}$ the $s$th moment of a $N(0,1)$ distributed random variable. Let $q$ be any number satisfying $p \leq q \leq 2$. Introduce the function $f_q(\mathbf{w}) := \frac{1}{2}\|\mathbf{w}\|_{2,q}^2$. Any $h^{\mathbf{w}} \in H_{q,\Lambda}$ satisfies the inequality

$$f_q(\mathbf{w}) = \frac{1}{2}\|\mathbf{w}\|_{2,q}^2 \leq \frac{1}{2}\Lambda^2.$$

Since $f_q(\mathbf{w})$ is $1/q^*$-strongly convex w.r.t. the norm $\|\cdot\|_{2,q}$, and the dual norm of $\|\cdot\|_{2,q}$ is $\|\cdot\|_{2,q^*}$ (Cf. section 4.2 in [8]), the summation of the squared dual norm in Theorem 7 can be rewritten as

follows:

$$\sum_{i=1}^{n} \mathbb{E}_{\boldsymbol{g}} \|(g_i\phi(x_i), \ldots, g_{(c-1)n+i}\phi(x_i))\|_{2,q^*}^2 = \sum_{i=1}^{n} \mathbb{E}_{\boldsymbol{g}} \Big[ \sum_{j=1}^{c} \|g_{(j-1)n+i}\phi(x_i)\|_2^{q^*} \Big]^{\frac{2}{q^*}}$$

$$= \sum_{i=1}^{n} \mathbb{E}_{\boldsymbol{g}} \Big[ \sum_{j=1}^{c} |g_{(j-1)n+i}|^{q^*} \Big]^{\frac{2}{q^*}} k(x_i, x_i)$$

$$\overset{\text{symmetry}}{=} \mathbb{E}_{\boldsymbol{g}} \Big[ \sum_{j=1}^{c} |g_j|^{q^*} \Big]^{\frac{2}{q^*}} \sum_{i=1}^{n} k(x_i, x_i)$$

$$\overset{\text{Jensen}}{\leq} c^{\frac{2}{q^*}} \tau_{q^*}^2 \sum_{i=1}^{n} k(x_i, x_i).$$

From which Theorem 7 immediately implies the following bounds, with probability at least $1 - \delta$ and for any $h^{\mathbf{w}} \in H_{q,\Lambda}$:

$$R(h^{\mathbf{w}}) \leq \frac{1}{n} \sum_{i=1}^{n} \ell(\rho_{h^{\mathbf{w}}}(x_i, y_i)) + \frac{4L\Lambda c^{1/q^*}\tau_{q^*}}{n} \sqrt{\frac{\pi q^*}{2} \sum_{i=1}^{n} k(x_i, x_i)} + 3B_\ell \sqrt{\frac{\log\frac{2}{\delta}}{2n}}.$$

From the trivial inequality $\|\mathbf{w}\|_{2,p} \geq \|\mathbf{w}\|_{2,q}$, we immediately conclude $H_{p,\Lambda} \subset H_{q,\Lambda}$. Therefore, for any $h^{\mathbf{w}} \in H_{p,\Lambda}$, we have

$$R(h^{\mathbf{w}}) \leq \frac{1}{n} \sum_{i=1}^{n} \ell(\rho_{h^{\mathbf{w}}}(x_i, y_i)) + \inf_{p \leq q \leq 2} \frac{4L\Lambda c^{1/q^*}\tau_{q^*}}{n} \sqrt{\frac{\pi q^*}{2} \sum_{i=1}^{n} k(x_i, x_i)} + 3B_\ell \sqrt{\frac{\log\frac{2}{\delta}}{2n}}.$$

It can be directly checked that the function $t \to \sqrt{t}c^{1/t}$ is decreasing along the interval $(0, 2\log c)$ and increasing along the interval $(2\log c, \infty)$. Therefore, the above generalization bound satisfies the inequality

$$R(h^{\mathbf{w}}) \leq \frac{1}{n} \sum_{i=1}^{n} \ell(\rho_{h^{\mathbf{w}}}(x_i, y_i)) + 3B_\ell \sqrt{\frac{\log\frac{2}{\delta}}{2n}} +$$

$$\frac{L\Lambda}{n} \sqrt{8 \sum_{i=1}^{n} k(x_i, x_i)} \times \begin{cases} \sqrt{2e\log c}\,\tau_{2\log c}, & \text{if } p^* \geq 2\log c, \\ c^{\frac{1}{p^*}}\tau_{p^*}\sqrt{p^*}, & \text{otherwise.} \end{cases}$$

Applying Lemma B.3 to bound the moments of Gaussian variables, the stated result follows immediately. $\qquad\square$

## C  Proofs on the Dual Problems

### C.1  Equivalent Representation of $\ell_p$-norm Multi-class Classification

The equivalence between Problem (P) and Eq. (8) follows directly from the following lemma due to [9].

**Lemma C.1** ([9]). *Let* $a_i \geq 0, i \in \mathbb{N}_d$ *and* $1 \leq r < \infty$. *Then*

$$\min_{\eta:\eta_i \geq 0, \sum_{i \in \mathbb{N}_d} \eta_i^r \leq 1} \sum_{i \in \mathbb{N}_d} \frac{a_i}{\eta_i} = \left( \sum_{i \in \mathbb{N}_d} a_i^{\frac{r}{r+1}} \right)^{1+\frac{1}{r}}$$

*and the minimum is attained at*

$$\eta_i = \frac{a_i^{\frac{1}{r+1}}}{\left( \sum_{k \in \mathbb{N}_d} a_k^{\frac{r}{r+1}} \right)^{\frac{1}{r}}}.$$

**Proof of Proposition 16**. Fixing $\mathbf{w}$, the sub-optimization of Eq. (8) w.r.t. $\boldsymbol{\beta}$ is

$$\min_{\boldsymbol{\beta}} \sum_{j=1}^{c} \frac{\|\mathbf{w}_j\|_2^2}{2\beta_j}$$

$$\text{s.t. } \|\boldsymbol{\beta}\|_{\bar{p}} \leq 1, \bar{p} = p(2-p)^{-1}, \beta_j \geq 0.$$

The stated result now follows directly by applying Lemma C.1 with $r = \bar{p}$ and $\alpha_j = \|\mathbf{w}_j\|_2^2$. $\quad\square$

## C.2 Derivation of the Completely Dualized Problem (Problem 11)

**Derivation of Problem 11**. Problem (P) translates to the following equivalent problem

$$\min_{\mathbf{w}} \frac{1}{2} \Big[ \sum_{j=1}^{c} \|\mathbf{w}_j\|_2^p \Big]^{\frac{2}{p}} + C \sum_{i=1}^{n} \ell(t_i) \tag{C.1}$$

$$\text{s.t. } t_i \leq \langle \mathbf{w}_{y_i}, \phi(x_i) \rangle - \langle \mathbf{w}_y, \phi(x_i) \rangle, \quad y \neq y_i, i = 1, \ldots, n.$$

The Lagrangian of the above convex optimization problem is

$$\mathcal{L} = \frac{1}{2} \Big[ \sum_{j=1}^{c} \|\mathbf{w}_j\|_2^p \Big]^{\frac{2}{p}} + C \sum_{i=1}^{n} \ell(t_i) + \sum_{i=1}^{n} \sum_{j \neq y_i} \tilde{\alpha}_{ij} \big( t_i + \langle \mathbf{w}_j, \phi(x_i) \rangle - \langle \mathbf{w}_{y_i}, \phi(x_i) \rangle \big),$$

with Lagrangian variables $0 \leq \tilde{\boldsymbol{\alpha}} \in \mathbb{R}^{n \times (c-1)}$. For the last term of the Lagrangian, we have the following identity:

$$\sum_{i=1}^{n} \sum_{j \neq y_i} \tilde{\alpha}_{ij} \langle \mathbf{w}_j - \mathbf{w}_{y_i}, \phi(x_i) \rangle = \sum_{i=1}^{n} \sum_{j \neq y_i} \tilde{\alpha}_{ij} \langle \mathbf{w}_j, \phi(x_i) \rangle - \sum_{i=1}^{n} \sum_{\tilde{j} \neq y_i} \tilde{\alpha}_{i\tilde{j}} \langle \mathbf{w}_{y_i}, \phi(x_i) \rangle$$

$$= \sum_{j=1}^{c} \langle \mathbf{w}_j, \sum_{i:y_i \neq j} \tilde{\alpha}_{ij} \phi(x_i) \rangle - \sum_{j=1}^{c} \sum_{i:y_i=j} \sum_{\tilde{j} \neq j} \tilde{\alpha}_{i\tilde{j}} \langle \mathbf{w}_j, \phi(x_i) \rangle \tag{C.2}$$

$$= \sum_{j=1}^{c} \langle \mathbf{w}_j, \sum_{i:y_i \neq j} \tilde{\alpha}_{ij} \phi(x_i) - \sum_{i:y_i=j} \sum_{\tilde{j} \neq j} \tilde{\alpha}_{i\tilde{j}} \phi(x_i) \rangle.$$

With this identity, the Lagrangian translates to

$$\mathcal{L} = \frac{1}{2} \Big[ \sum_{j=1}^{c} \|\mathbf{w}_j\|_2^p \Big]^{\frac{2}{p}} + \sum_{j=1}^{c} \langle \mathbf{w}_j, \sum_{i:y_i \neq j} \tilde{\alpha}_{ij} \phi(x_i) - \sum_{i:y_i=j} \sum_{\tilde{j} \neq j} \tilde{\alpha}_{i\tilde{j}} \phi(x_i) \rangle +$$

$$C \sum_{i=1}^{n} [\ell(t_i) + \frac{1}{C} \sum_{\tilde{j} \neq y_i} \tilde{\alpha}_{i\tilde{j}} t_i]. \tag{C.3}$$

According to the definition of Fenchel conjugate function, it holds that

$$\inf_{\mathbf{w},\mathbf{t}} \mathcal{L} = -\sup_{\mathbf{w}} \Big[ -\frac{1}{2} \Big[ \sum_{j=1}^{c} \|\mathbf{w}_j\|_2^p \Big]^{\frac{2}{p}} - \sum_{j=1}^{c} \langle \mathbf{w}_j, \sum_{i:y_i \neq j} \tilde{\alpha}_{ij} \phi(x_i) - \sum_{i:y_i=j} \sum_{\tilde{j} \neq j} \tilde{\alpha}_{i\tilde{j}} \phi(x_i) \rangle \Big]$$

$$- C \sum_{i=1}^{n} \sup_{t_i} [-\ell(t_i) - \sum_{j \neq y_i} \frac{1}{C} \tilde{\alpha}_{ij} t_i]$$

$$= -\Big[ \frac{1}{2} \Big\| \Big( -\sum_{i:y_i \neq j} \tilde{\alpha}_{ij} \phi(x_i) + \sum_{i:y_i=j} \sum_{\tilde{j} \neq j} \tilde{\alpha}_{i\tilde{j}} \phi(x_i) \Big)_{j=1}^{c} \Big\|_{2,p}^2 \Big]^*$$

$$- C \sum_{i=1}^{n} \ell^* \big( -\frac{1}{C} \sum_{j \neq y_i} \tilde{\alpha}_{ij} \big)$$

$$= -\frac{1}{2} \Big\| \Big( \sum_{i:y_i \neq j} \tilde{\alpha}_{ij} \phi(x_i) - \sum_{i:y_i=j} \sum_{\tilde{j} \neq j} \tilde{\alpha}_{i\tilde{j}} \phi(x_i) \Big)_{j=1}^{c} \Big\|_{2,\frac{p}{p-1}}^2 - C \sum_{i=1}^{n} \ell^* \big( -\frac{1}{C} \sum_{j \neq y_i} \tilde{\alpha}_{ij} \big),$$

$$\tag{C.4}$$

where in the last step of the above deduction we have used the identity: $\left(\frac{1}{2}\|\cdot\|^2\right)^* = \frac{1}{2}\|\cdot\|_*^2$ and the fact that the dual norm of $\|\cdot\|_{2,p}$ is $\|\cdot\|_{2,\frac{p}{p-1}}$. Consequently, the dual problem becomes

$$\sup_{\tilde{\boldsymbol{\alpha}}\in\mathbb{R}^{n\times(c-1)}} -\frac{1}{2}\Big[\sum_{j=1}^{c}\|\sum_{i:y_i\neq j}\tilde{\alpha}_{ij}\phi(x_i) - \sum_{i:y_i=j}\sum_{\tilde{j}\neq j}\tilde{\alpha}_{i\tilde{j}}\phi(x_i)\|_2^{\frac{p}{p-1}}\Big]^{\frac{2(p-1)}{p}} - C\sum_{i=1}^{n}\ell^*\big(-\frac{1}{C}\sum_{j\neq y_i}\tilde{\alpha}_{ij}\big),$$

$$\text{s.t. } \tilde{\boldsymbol{\alpha}}\geq 0.$$

Introducing $\boldsymbol{\alpha}\in\mathbb{R}^{n\times c}$ via the substitution:

$$\alpha_{ij} = \begin{cases} -\tilde{\alpha}_{ij} & \text{if } j\neq y_i \\ \sum_{\tilde{j}\neq y_i}\tilde{\alpha}_{i\tilde{j}} & \text{if } j=y_i, \end{cases} \tag{C.5}$$

we have

$$\sum_{i:y_i\neq j}\tilde{\alpha}_{ij}\phi(x_i) - \sum_{i:y_i=j}\sum_{\tilde{j}\neq j}\tilde{\alpha}_{i\tilde{j}}\phi(x_i) = -\sum_{i:y_i\neq j}\alpha_{ij}\phi(x_i) - \sum_{i:y_i=j}\alpha_{ij}\phi(x_i), \tag{C.6}$$

from which the stated dual problem follows directly. $\qquad\square$

### C.3 Proof of the Representer Theorem (Theorem 12)

Let $H_1,\ldots,H_c$ be $c$ Hilbert spaces and $p\geq 1$. Define the function $g_p(v_1,\ldots,v_c): H_1\times\cdots\times H_c \to \mathbb{R}$ by

$$g_p(v_1,\ldots,v_c) = \frac{1}{2}\|(v_1,\ldots,v_c)\|_{2,p}^2, \quad p\geq 1.$$

**Lemma C.2.** *The gradient of $g_p$ is*

$$\frac{\partial g_p(v_1,\ldots,v_c)}{\partial v_j} = \Big[\sum_{\tilde{j}=1}^{c}\|v_{\tilde{j}}\|_2^p\Big]^{\frac{2}{p}-1}\|v_j\|_2^{p-2}v_j.$$

*Proof.* By the chain rule, we have

$$\frac{\partial g_p(v_1,\ldots,v_c)}{\partial v_j} = \frac{1}{p}\Big[\sum_{\tilde{j}=1}^{c}\|v_{\tilde{j}}\|_2^p\Big]^{\frac{2}{p}-1}\frac{\partial\langle v_j,v_j\rangle^{\frac{p}{2}}}{\partial v_j}$$

$$= \frac{1}{2}\Big[\sum_{\tilde{j}=1}^{c}\|v_{\tilde{j}}\|_2^p\Big]^{\frac{2}{p}-1}\frac{\partial\langle v_j,v_j\rangle}{\partial v_j}\langle v_j,v_j\rangle^{\frac{p}{2}-1}$$

$$= \Big[\sum_{\tilde{j}=1}^{c}\|v_{\tilde{j}}\|_2^p\Big]^{\frac{2}{p}-1}\|v_j\|_2^{p-2}v_j.$$

$\qquad\square$

**Proof of Representer Theorem (Theorem 12)**. In our derivation of the dual problem (see Eq. (C.4)), the variable $\mathbf{w}$ should meet the optimality in the sense that

$$\mathbf{w} = \arg\max_{\boldsymbol{v}} -\frac{1}{2}\Big[\sum_{j=1}^{c}\|\boldsymbol{v}_j\|_2^p\Big]^{\frac{2}{p}} + \sum_{j=1}^{c}\langle\boldsymbol{v}_j, \sum_{i=1}^{n}\alpha_{ij}\phi(x_i)\rangle.$$

Since $(\bigtriangledown f)^{-1} = \bigtriangledown f^*$ for any convex function $f$, and the Fenchel-conjugate of $g_p$ is $g_{p^*}$, we obtain the following representation of $\mathbf{w}$:

$$\mathbf{w} = \bigtriangledown g_p^{-1}\Big(\sum_{i=1}^{n}\alpha_{i1}\phi(x_i),\ldots,\sum_{i=1}^{n}\alpha_{ic}\phi(x_i)\Big)$$

$$= \bigtriangledown g_{p^*}\Big(\sum_{i=1}^{n}\alpha_{i1}\phi(x_i),\ldots,\sum_{i=1}^{n}\alpha_{ic}\phi(x_i)\Big)$$

$$= \Big[\sum_{j=1}^{c}\|\sum_{i=1}^{n}\alpha_{ij}\phi(x_i)\|_2^{p^*}\Big]^{\frac{2}{p^*}-1}\Big(\|\sum_{i=1}^{n}\alpha_{i1}\phi(x_i)\|_2^{p^*-2}\big[\sum_{i=1}^{n}\alpha_{i1}\phi(x_i)\big],\ldots\|\sum_{i=1}^{n}\alpha_{ic}\phi(x_i)\|_2^{p^*-2}\big[\sum_{i=1}^{n}\alpha_{ic}\phi(x_i)\big]\Big).$$

That is,

$$\mathbf{w}_j = \Big[\sum_{\tilde{j}=1}^{c} \|\sum_{i=1}^{n} \alpha_{i\tilde{j}}\phi(x_i)\|_2^{p^*}\Big]^{\frac{2}{p^*}-1} \|\sum_{i=1}^{n} \alpha_{ij}\phi(x_i)\|_2^{p^*-2} \Big[\sum_{i=1}^{n} \alpha_{ij}\phi(x_i)\Big].$$

$\square$

### C.4 Derivation of Partially Dualized Problem (Problem 14)

**Derivation of Problem 14**. The Lagrangian of the problem (8) w.r.t. $\mathbf{w}$ is

$$\mathcal{L} = \sum_{j=1}^{c} \frac{\|\mathbf{w}_j\|_2^2}{2\beta_j} + C\sum_{i=1}^{n} \ell(t_i) + \sum_{i=1}^{n}\sum_{j\neq y_i} \tilde{\alpha}_{ij}\big(t_i + \langle \mathbf{w}_j, \phi(x_i)\rangle - \langle \mathbf{w}_{y_i}, \phi(x_i)\rangle\big),$$

with Lagrangian variables $0 \leq \tilde{\boldsymbol{\alpha}} \in \mathbb{R}^{n\times(c-1)}$.

According to the identity (C.2), the Lagrangian translates to

$$\mathcal{L} = \sum_{j=1}^{c} \frac{\|\mathbf{w}_j\|_2^2}{2\beta_j} + \sum_{j=1}^{c}\langle \mathbf{w}_j, \sum_{i:y_i\neq j}\tilde{\alpha}_{ij}\phi(x_i) - \sum_{i:y_i=j}\sum_{\tilde{j}\neq j}\tilde{\alpha}_{i\tilde{j}}\phi(x_i)\rangle +$$

$$C\sum_{i=1}^{n}[\ell(t_i) + \frac{1}{C}\sum_{\tilde{j}\neq y_i}\tilde{\alpha}_{i\tilde{j}}t_i]. \quad \text{(C.7)}$$

According to the definition of Fenchel conjugate function, it holds that

$$\inf_{\mathbf{w},\mathbf{t}} \mathcal{L} = -\sum_{j=1}^{c}\Big[\frac{1}{\beta_j}\sup_{\mathbf{w}_j}\big[-\frac{1}{2}\|\mathbf{w}_j\|_2^2 - \langle \mathbf{w}_j, \beta_j\big(\sum_{i:y_i\neq j}\tilde{\alpha}_{ij}\phi(x_i) - \sum_{i:y_i=j}\sum_{\tilde{j}\neq j}\tilde{\alpha}_{i\tilde{j}}\phi(x_i))\rangle\big)\big]\Big]$$

$$- C\sum_{i=1}^{n}\sup_{t_i}[-\ell(t_i) - \sum_{j\neq y_i}\frac{1}{C}\tilde{\alpha}_{ij}t_i]$$

$$= -\sum_{j=1}^{c}\Big[\frac{1}{\beta_j}\big[\frac{1}{2}\|\beta_j\big(\sum_{i:y_i\neq j}\tilde{\alpha}_{ij}\phi(x_i) - \sum_{i:y_i=j}\sum_{\tilde{j}\neq j}\tilde{\alpha}_{i\tilde{j}}\phi(x_i))\|_2^2\big]^{*}\big]\Big] - C\sum_{i=1}^{n}\ell^{*}\big(-\frac{1}{C}\sum_{j\neq y_i}\tilde{\alpha}_{ij}\big)$$

$$= -\frac{1}{2}\sum_{j=1}^{c}\beta_j\Big\|\sum_{i:y_i\neq j}\tilde{\alpha}_{ij}\phi(x_i) - \sum_{i:y_i=j}\sum_{\tilde{j}\neq j}\tilde{\alpha}_{i\tilde{j}}\phi(x_i)\Big\|_2^2 - C\sum_{i=1}^{n}\ell^{*}\big(-\frac{1}{C}\sum_{j\neq y_i}\tilde{\alpha}_{ij}\big),$$

where in the last step of the above deduction we have used the identity: $\big(\frac{1}{2}\|\cdot\|^2\big)^{*} = \frac{1}{2}\|\cdot\|_*^2$ and the fact that the dual norm of $\|\cdot\|_{2,2}$ is itself. Consequently, the dual problem becomes

$$\sup_{\tilde{\boldsymbol{\alpha}}\in\mathbb{R}^{n\times(c-1)}} -\frac{1}{2}\sum_{j=1}^{c}\beta_j\Big\|\sum_{i:y_i\neq j}\tilde{\alpha}_{ij}\phi(x_i) - \sum_{i:y_i=j}\sum_{\tilde{j}\neq j}\tilde{\alpha}_{i\tilde{j}}\phi(x_i)\Big\|_2^2 - C\sum_{i=1}^{n}\ell^{*}\big(-\frac{1}{C}\sum_{j\neq y_i}\tilde{\alpha}_{ij}\big),$$

$$\text{s.t. } \tilde{\boldsymbol{\alpha}} \geq 0.$$

Introducing $\boldsymbol{\alpha} \in \mathbb{R}^{n\times c}$ as in Eq. (C.5) and noticing the identity (C.6), the above *dual problem* becomes

$$\sup_{\boldsymbol{\alpha}\in\mathbb{R}^{n\times c}} -\frac{1}{2}\sum_{j=1}^{c}\beta_j\Big\|\sum_{i=1}^{n}\alpha_{ij}\phi(x_i)\Big\|_2^2 - C\sum_{i=1}^{n}\ell^{*}(-\frac{\alpha_{iy_i}}{C})$$

$$\text{s.t. } \sum_{j=1}^{c}\alpha_{ij} = 0, \quad \forall i = 1, 2, \ldots, n, \quad \text{(C.8)}$$

$$\alpha_{ij} \leq 0, \qquad j \neq y_i, \forall i = 1, \ldots, n.$$

Note that in the above derivation of the dual problem, the variable $\mathbf{w}$ should meet the optimality in the sense that

$$\mathbf{w} = \arg\max_{\boldsymbol{v}} -\frac{1}{2}\sum_{j=1}^{c}\|\boldsymbol{v}_j\|_2^2 + \sum_{j=1}^{c}\beta_j\langle \boldsymbol{v}_j, \sum_{i=1}^{n}\alpha_{ij}\phi(x_i)\rangle.$$

The representer theorem stated in Problem 14 follows directly from this optimization condition. $\square$

## D   Note on Used Features for Caltech256 and UCSD birds

For the purpose of having features, we took the features from a fc6 layer of the BVLC reference caffenet [10] computed for all images from the UCSD birds dataset [11] and Caltech256 [12]. Note that we neither used fc7 or fc8 layers, nor did we perform finetuning. Images were warped [13] so that they fitted into the quadratic reception field. As the goal was not on maximizing performance but comparing learning machines we resorted to computing one feature per image at training and test time without using the large number of region proposals which yield state of the art in fine-grained classification tasks [14], or mirroring and detection-like approaches like the 500 windows per image as in [15].

## E   Execution Time Experiments

This section report the training time of the classical CS [16] and the proposed $\ell_p$-norm MC-SVM on the benchmark datasets. We repeat the experiments 10 times and report the average as well as standard deviation (in seconds) in Table E.1. For our method, the result is for the single $p$ selected via cross-validation.

| Method / Dataset | Sector | News 20 | Birds 15 | Birds 50 |
|---|---|---|---|---|
| $\ell_p$-norm MC-SVM | $4914 \pm 64.7$ | $3894 \pm 71.1$ | $912.6 \pm 22.1$ | $518.4 \pm 34.8$ |
| Crammer & Singer | $3442 \pm 91.8$ | $2227 \pm 43.7$ | $701.7 \pm 50.6$ | $314.1 \pm 17.1$ |

Table E.1: Training time for the classical CS and the proposed $\ell_p$-norm MC-SVM on the benchmark datasets.

From Table E.1, we see that our method needs longer training time than CS, but the increase is not that large and is well compensated by its improvement on accuracies.