[Reviews · NeurIPS 2015]

Submitted by Assigned_Reviewer_1

The paper describes a family of multi-class SVMs parameterised by a single parameter p. The family contains the Crammer & Singer SVM as a special case.

Corresponding generalization bounds are derived, which for certain parameter choices have a logarithmic dependency on the number of classes. The latter has - to my knowledge - not been shown for any multi-class classifier.

The corresponding learning machine is derived including its Fenchel dual problem. There is an empirical evaluation, which shows that the approach works. However, the experiments are the weak spot of the paper.

Only three test problems are considered. It is not clear which kernel function is used, therefore I assume a linear kernel. The Crammer & Singer SVM serves as the baseline, the parameter p is optimised on a fine grid (including the value p=2, which gives the Crammer & Singer SVM). The results for the best p value are reported (optimised for each data set independently), which are slightly better than the Crammer & Singer SVM results.

That the Crammer & Singer SVM results are achieved does not come as a surprise, as p=2 is tested for the new machines, i.e., the new machines include the Crammer & Singer SVM. The grid for optimising p is very fine. That is, the best of many values of p (i.e., many different machines) are compared to the Crammer & Singer variant. That one always find one machine for each data set is not surprising, actually I think that the improvements are so small is rather disappointing. What if the model selection for the Crammer & Singer variant is refined such that in the end as many machines are tested as in the case when also p is optimised? For example, using a finer grid for regularisation parameter and kernel bandwidth parameter (in case of a Gaussian kernel SVM). Better statistics about the robustness w.r.t. p should be reported.

Testing more data sets with a linear as well as a non-linear kernel is mandatory.

Computation times for solving the optimisation problems are not given. The general problem of the new machines is more difficult to solve than the Crammer & Singer QP, and the scaling of the optimisation time w.r.t. the number of training data points is important to see. Timing experiments are a must.

Long training times (which I assume) and the necessity to adapt an additional hyperparameter reduce the practical value of the proposed approach.

Anyway, the theoretical result is strong.

Minor comments: mercer -> Mercer lines 394,395: The description of the grid for p does not seem to be consistent. 1.2 and 1.25 differ by 0.5 - how is this related to the step sizes 0.5 and 0.1?
Summary: New family of multi-class SVMS, which comes with strong generalisation guarantees. The theoretical result is strong, the empirical evaluation is not convincing.

Submitted by Assigned_Reviewer_2

The paper presents a novel data-dependent learning bound (theorem 7) for multi-class classification. The key highlights of this bound are: i) it is generic in the sense that it applies to a general family of regularizers ii) Corollary 8 shows e.g. of a nice family of regularizers where the dependence on no. classes is mild (grows like \sqrt{c} for multi-class SVMs i.e., p=2).

Here are some comments:

1. The bound is interesting for the two reasons mentioned above.

2. Though theorem 7 is general, the entire development later focused on lp-norm regularizer. This seems to provide restricted insights/improvements: a. In this case, logarithmic dependence is possible only if $p$ is close to 1. However, p close to 1 might boost up the empirical error term owing to sparsity in w. b. The above seems to manifest itself in the simulations that indeed show only marginal improvements. It would be interesting to infact note what were the actual tuned values of p in the simulations? were they really close to 1 or 10? c. The resulting formulation (Prob. 10) itself is not novel (and thus the algorithm), because with appropriate definition of kernel (over X\timesY) and loss function, it is a special case of l-p-MKL.

In summary, exploration of other potentially interesting regularizers may improve the bound, performance, as well as novelty in the formulation/algorithm.

3. The proof of Lemma 4 seems to apply Lemma A.1 with the index set as H. However H need not be a countable set, which seems to be a requirement for Lemma A.1. Is this an issue? or am I missing something?

4. Perhaps another baseline for comparison in simulations could be [18] (which also presents a novel formulation based on learning bounds).

5. Given that the accuracies are so close, a statistical significance test would have helped.
Summary: Overall, the paper is very well written and presents an interesting, generic data-dependent bound derived. However, further exploration of regularizers other than p-norm would have improved the paper substantially.

Submitted by Assigned_Reviewer_3

The main result is the theoretical analysis that is based on the insight that coupling effects across classes may be exploited to obtain better bounds. The part explaining the coupling and its consequences (e.g., Eq 3 to Eq 4) could be more detailed as it lays the ground for the remainder. Generally, the analysis is well written and a discussion with related work allows to differentiate the contribution from existing approaches.

What follows is an elegant but straight forward derivation of a generalisation of the Crammer & Singer formulation using p-norms instead of the usual 2-norm regularisation.

The empirical results are computed on three pretty outdated and small data sets. The improvements are minor but significant, however, only the Crammer & Singer approach is introduced as a competitor. Although most of the multi-class classification approaches perform comparably, the set of baselines should be extended to showcase the variance of outcomes across several approaches.

Line 201 contains a qed box that should not be there.

Summary: The authors provide new theoretical insights into multi-class classification and, based on their observations, derive a a generalisation of Crammer and Singer (2002).

Submitted by Assigned_Reviewer_4

This is a heavily theoretical paper on the rate of convergence for multi-class SVM. The state-of-art of the rate of convergence is linearly dependent on the class size. This paper proves it to sub-linear dependence. In certain special case, the rate of convergence is in log(c). Overall, the paper is well written and has some originality. The algorithm has some potential.

However, log(c) is actually not significantly smaller than c (or phrased differently, the improvement is not good enough), because usually the class size c is fixed. Even in the simulation study and data examples, the class size is not that big.
Summary: The improvement from linear dependence to sublinear dependence on the class size does not seem to be significant. For example, when c=20 (20 classes), the difference between 20 and log(20) is actually not big. What is more important is the rate of convergence in $n$.

Author Feedback
Author rebuttal: We thank the reviewers for their constructive comments, and find it encouraging that the reviewers agree on the importance of the work.

[Rev 2, Rev 4]
* expect more datasets & more baseline algorithms
- Due to time limitations for rebuttal, we focused on producing results for more datasets. Here are the results (average accuracies pm stds) on new datasets:

(c=No. of classes, n=No. of training examples, d=No. of features, CS=Crammer & Singer)

Birds 15 (c=200, n=3000, d=4097) CS: 12.53 pm 1.6, Ours: 13.73 pm 1.4
Birds 50 (c=200, n=9958, d=4097) CS: 26.28 pm 0.3, Ours: 27.86 pm 0.2
Caltech 256 (c=256, n=12800,d=4097)CS: 54.96 pm 1.1, Ours: 56.00 pm 1.2

Results indicate that our method work consistently better than CS. We also agree that considering more baselines is very beneficial and plan to add more baseline comparisons for the camera-ready version.

[Rev 2]
* advises detailed explanation of coupling
- This is a nice suggestion and we will explain it in detail in the final version.

[Rev 3]
* worries on result's significance
- Indeed, the experiments show that the lp-norm regularizer yields an advantage also in problems where the class size does not matter such as news20 and rcv1. A bound with mild dependence on class size is important to problems with huge classes, of which we mentioned some examples in line 30. Another significance is its potential for application to structured prediction. Also, Theorem 7 applies to general regularizers and could motivate & explain new algorithms.

* worries on small class-size datasets
- We added datasets with larger class-size: Birds 15, Birds 50, Caltech 256.

[Rev 4]
* query on the kernel type & advice on non-linear kernel
- Yes, we use linear kernel (line 323). Non-linear kernels, which we didn't consider as SDM in [25] is not applicable, are interesting and will be our research direction.

* advices on model selection
- This is worth for a longer version paper. From our experience with experiments we did not observe a strong sensitivity of CS on the regularization constant.

* query on statistics on the robustness w.r.t p
- This is a nice remark to understand the influence of p. From our experience on experiments we observed that the validated p varies with different datasets. We will study this problem in more detail for a longer version paper.

* asks for timing experiment
- We added some timing experiments. Here are the results (average pm stds) in seconds (for our method, the result is for the single p selected via cross-validation):

Birds 15 --- CS: 701.7 pm 50.6, Ours: 912.6 pm 22.1
Birds 50 --- CS: 314.1 pm 17.1, Ours: 518.4 pm 34.8
Sector --- CS: 3442 pm 91.8, Ours: 4914 pm 64.7
News20 --- CS: 2227 pm 43.7, Ours: 3894 pm 71.1

We see that our method needs longer training time, but the increase is not that large and is well compensated by its improvement on accuracies.
Due to the time limit we do not have timing results on all datasets. We will include them in the final version.

* query on the grid for p
- Thanks for the query. We feel very sorry that the grid for p should be from 1.1 to 2, with 10 equidistant points.

[Rev 5, Rev 6]
Thank you for your positive reviews.

[Rev 7]
* suggests an exploration on regularizers
- Thank you for sharing the research idea with us, which we plan to explore in the upcoming research. Where we plan to consider a multi-task regularizer by encoding prior information among classes.

* query on the tuned values of p
- The validated p is dataset-dependent. In our experiments, it is in the interval [1.4,1.8]. We will report these results.

* remarks the lack of novelty in Prob 10
- We agree Prob 10 is motivated by our theoretical analysis and recent work on lp-MKL. However, Prob 10 differs from lp-MKL in: (1) Prob 10 only involves a single kernel; (2) the p-norm regularizer is enforced on classifiers with class-wise components, not on kernel-wise components in lp-MKL.

* query on proof of Lemma 4
- Lemma A.1 also holds when GPs are indexed by a separable space (a space with a countable dense subset), by monotone convergence theorem. GPs defined in proving Lemma 4 are indeed indexed by a finite dimensional Euclidean space (the projection of functions onto the sample), which is separable. We will clarify this in the paper.

* expects a comparison to M^3K [18]
- We agree that M^3K could serve as another baseline, but we could not produce these baseline experiments given the short time for the rebuttal (planed for camera-ready version). Meanwhile note, however, that there is a difference or a potential disadvantage of the M^3K approach in comparison with ours: M^3K is a MKL method, while ours is a single kernel method.

* expects a significance test
- We did a Wilcoxon signed rank test, and the p-value is 0.031, which means our method is significantly better than CS at the significance level of 0.05. We will include this result in the paper.